# Mass Timber Building Life Cycle Assessment Methodology for the U.S. Regional Case Studies

**Hongmei Gu [1,\*]** , **Shaobo Liang [1]** , **Francesca Pierobon [2]** , **Maureen Puettmann [3]** , **Indroneil Ganguly [2]** , **Cindy Chen [4]** , **Rachel Pasternack [5]** , **Mark Wishnie [6]** , **Susan Jones [7]** and **Ian Maples [7]**

1. Forest Products Laboratory, USDA Forest Service, Madison, WI 53726, USA; sshliang@gmail.com
2. School of Environmental and Forest Science, University of Washington, Seattle, WA 98105, USA; pierobon@uw.edu (F.P.); indro@uw.edu (I.G.)
3. CORRIM—The Consortium for Research on Renewable Industrial Materials, Corvallis, OR 97339, USA; maureen@corrim.org
4. Population Research Center, Portland State University, Portland, OR 97207, USA; cxc2@pdx.edu
5. The Nature Conservancy, Arlington, VA 22203, USA; Rachel.pasternack@TNC.org
6. BTG Pactual Timberland Investment Group, LLC, Seattle, WA 98199, USA; mark.wishnie@btgpactual.com
7. atelierjones, Seattle, WA 98101, USA; Susan@atelierjones.com (S.J.); ian@atelierjones.com (I.M.)
\* Correspondence: Hongmei.gu@usda.gov; Tel.: +1-608-231-9589

**Abstract:** The building industry currently consumes over a third of energy produced and emits 39% of greenhouse gases globally produced by human activities. The manufacturing of building materials and the construction of buildings make up 11% of those emissions within the sector. Whole-building life-cycle assessment is a holistic and scientific tool to assess multiple environmental impacts with internationally accepted inventory databases. A comparison of the building life-cycle assessment results would help to select materials and designs to reduce total environmental impacts at the early planning stage for architects and developers, and to revise the building code to improve environmental performance. The Nature Conservancy convened a group of researchers and policymakers from governments and non-profit organizations with expertise across wood product life-cycle assessment, forest carbon, and forest products market analysis to address emissions and energy consumption associated with mass timber building solutions. The study disclosed a series of detailed, comparative life-cycle assessments of pairs of buildings using both mass timber and conventional materials. The methodologies used in this study are clearly laid out in this paper for transparency and accountability. A plethora of data exists on the favorable environmental performance of wood as a building material and energy source, and many opportunities appear for research to improve on current practices.

**Keywords:** cross-laminated timber; life-cycle assessment; mass timber building; whole-building LCA methodology

## 1. Introduction

The most recent Intergovernmental Panel on Climate Change (IPCC) report [1] has issued its starkest warning for immediate action in the next decade from every country and every industry to fight global temperature rise and prevent climate catastrophe. The building industry plays an essential role in tackling climate change because 36% of global energy use and 39% of energy-related carbon emissions are coming from the building sector. This is primarily due to the traditional use of fossil-based, carbon-intensive building materials such as concrete and steel. With new engineered wood products, called mass timber (e.g., cross-laminated timber (CLT)), being developed for mid- to high-rise buildings, the building sector has the potential to significantly lower associated carbon emissions [2–5]. Trees absorbing carbon from the atmosphere make forests a carbon sink, and mass timber buildings storing this carbon during their service life will change the building sector from a giant carbon emitter to a giant carbon tank [6–8].

Natural climate solutions through forest management, growing more trees, and using trees to make long-lived products can increase carbon storages or avoid greenhouse gas emissions. Research has indicated that natural climate solutions could deliver 30% or more of the mitigation required to achieve the goals of the Paris Agreement through 2030, with reforestation potentially offering the single greatest natural climate solution opportunity [9,10]. Mass timber products, such as CLT, glued-laminated timber (glulam), and dowel-laminated timber (DLT), are becoming the preferred products of architects, builders, and engineers because of their inherent low carbon footprint, long-term carbon storage, and renewable nature. Wood products from sustainably managed forests play an essential role in reducing carbon emissions in the building sector, where heavy petroleum or fossil-based materials have been used almost exclusively for decades [11].

To assess the potential impacts of greater forest products' utilization on global forests and climate, The Nature Conservancy initiated a global research program to examine the climate and forest impacts of the emerging mass timber use in buildings. The first phrase of this program is to develop a comparative life-cycle assessment (LCA) of functional equivalent mass timber and conventional buildings in selected U.S. regions to estimate embodied carbon and the carbon storage of mass timber utilization at the individual building level.

The LCA of a product, process, or system is an objective process to evaluate a product's life-cycle energy and material used, as well as the emissions and waste released, then assess the impacts of those inputs and outputs on the environment and implement strategies to reduce them. The whole-building LCA (WBLCA) uses the LCA approach to evaluate the building throughout its entire life, focusing not only on operational carbon, which refers to the impact of operating the building, but also on embodied carbon, which includes the impact of manufacturing the building, including material selection, material sourcing, and construction.

LCA is not only a computational tool to quantify the environmental impacts of products and processes, but also a comparative assessment tool, which requires LCA practitioners to follow best practices in making appropriate comparisons between products and materials [12]. With growing interest in applying LCA to whole-buildings, considerable variations have been discovered between case studies due to the methods, tools, and database used, leading to limitations in comparing results and conclusions that can be drawn. Kwok et al. [13] found significant differences in results between Tally (Tally® Life Cycle Assessment App) and the Athena Impact Estimator for Buildings (IE4B). These differences were a result of many variables, but the most significant were how each software calculated biogenic carbon and their methodologies to account for carbon storage and end-of-life scenarios. In addition, they do not always allow for transparency in the material choices and upstream production processes to allow for regional production differences.

Greater transparency and conformity in the methods and material datasets are needed for the WBLCA community to embrace results for a fair comparison and meaningful conclusion. International standards have been developed to guide WBLCA developers to ensure uniformity between material comparisons and buildings. The International Organization of Standardization (ISO 21931), together with the European Standard (EN 15978), are the fundamental standards followed when conducting WBLCAs. This paper lays out the WBLCA methodologies conducted following ISO 21931 and EN 15978 standards [14,15].

## 2. Methodology Development

### 2.1. Building Designs

The buildings were designed based on projected markets for new mass timber buildings. Both timber and concrete buildings were modeled with a mixed use of residential and office functions that would represent mass timber building use over the next 20 years. This assumption could be challenged by further research to analyze future urban building demand and a better understanding of future markets for tall wood building over the next two decades regarding their dominant program, be it office or residential. The decision

to develop the following programs (Table 1) were based on a professional judgement by the multidisciplinary team and was intended to approximate a mix of uses across regional building stock but could be improved through further analysis. Table 1 shows the building program for all three building heights, representing both mass timber and concrete buildings. Construction Type IV(A–C) applies only to mass timber buildings. Further details on building designs and program are provided in Appendix A of Maureen et al. [16]. All buildings were designed with mixed uses in mind, with double entries out to the sidewalks and double elevator cores. All building heights for both mass timber and concrete designs had the same footprint of 26 by 46 m (Figure 1).

**Table 1.** The designed mixed-use buildings for three levels—8, 12, and 18 story.

| Mass Timber Construction Type | Stories | Residential | Office/Commercial | Building Height | | Total Floor Area | |
|---|---|---|---|---|---|---|---|
| | | | | Feet | Meters | ft$^2$ | m$^2$ |
| Type IV-C | 8 | 6 stories | 2 stories | 85 | 26 | 102,000 | 9476 |
| Type IV-B | 12 | 8 stories | 4 stories | 156 | 48 | 153,000 | 14,214 |
| Type IV-A | 18 | 12 stories | 6 stories | 234 | 71 | 229,500 | 21,321 |

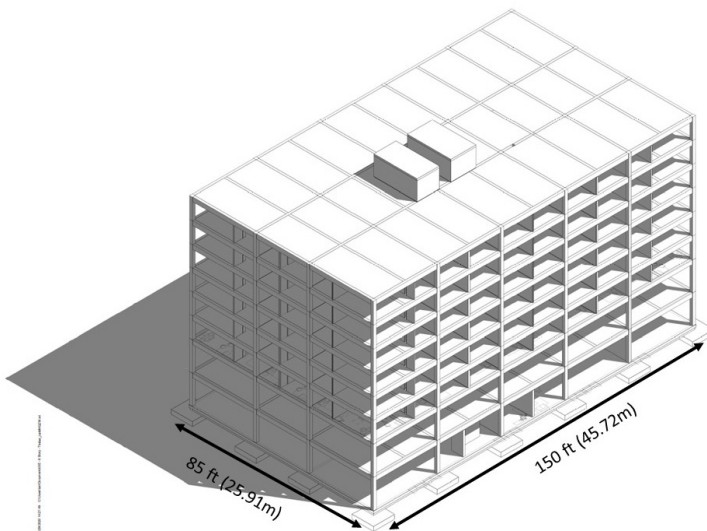

**Figure 1.** The mixed-use building design for this study.

Using designs and assumptions developed by the architecture company Atelierjones (atelierjones LLC), a comparative WBLCA's for mass timber and conventional concrete designs was developed. Eighteen different modeling conditions were selected for the comparative building LCAs in the United States (Figure 2). Each building design was hypothetically constructed in one of three U.S. geographic regions: 1. Seattle, WA, for the Pacific Northwest (PNW) region; 2. Boston, MA, for the Northeast (NE) region; 3. Atlanta, GA, for the Southeast (SE) region. Three pairs of model buildings were designed for three U.S. regions to conform to three tall mass timber building types established in the 2021 International Building Code (ICC) revisions for 8, 12, and 18 stories. The results allow for comparisons among regions, each of which has different energy mixes and timber species, and, in the case of the PNW, additional seismic considerations that drive differences in the building design. Conventional buildings were designed for functional equivalent comparisons with each of the mass timber buildings (e.g., equivalent height, interior floor space, interior divisions, and intended uses).

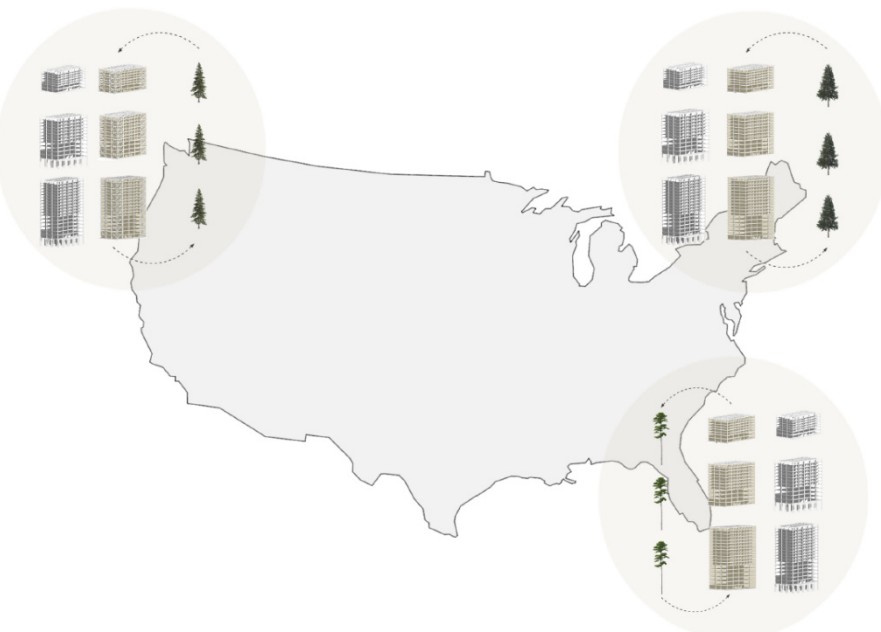

**Figure 2.** Buildings designed in three regions of the US–Pacific Northwest (Seattle, WA, USA), Northeast (Boston, MA, USA), and Southeast (Atlanta, GA, USA).

The concrete buildings utilize steel and concrete as the main materials. The mass timber buildings utilize steel and concrete in their foundations, elevator cores, and certain other structural elements, but maximize the use of mass timber in walls, floors, and other building elements. For all building designs in the PNW region, both mass timber and concrete buildings, the lateral system for each building had to be upgraded to account for a more robust lateral system due to the high seismic impact potential. The PNW timber building designs differed from the NE and SE designs because of this seismic requirement. Material quantities and choices were the same for the NE and SE mass timber and concrete buildings.

### 2.2. Bill of Materials Take-Off

All building designs were completed on the modeling software AutoDesk REVIT 2015 (Autodesk Revit 2015 Product Updates | Revit | Autodesk Knowledge Network). The plan templates were created for each building floor plan, including ground floor for typical office/commercial floor and residential floor. AutoDesk REVIT is a sophisticated Building Information Modeling program that has been a standard in the Architecture/Engineering/Contracting community for several decades. It is known for embedding specific construction materials into its database, so that, when designers model the building, specific materials can be easily quantified into automatically generated spreadsheets for contractors to use to establish quantities and budgets. The bill of materials used in these buildings are listed in Tables 2–4 for all 18 buildings.

**Table 2.** Bill of materials for 8-, 12-, and 18-story mass timber and concrete designs in the PNW.

| Assembly | Material: Name | Unit | 8-Story | | 12-Story | | 18-Story | |
|---|---|---|---|---|---|---|---|---|
| | | | Timber | Concrete | Timber | Concrete | Timber | Concrete |
| Columns and Beams | Concrete | m$^3$ | - | 93 | - | 385 | - | 586 |
| | Glulam | m$^3$ | 859 | - | 1376 | - | 2265 | - |
| | Rebar | | | | | | - | 285,609 |

**Table 2.** *Cont.*

| Assembly | Material: Name | Unit | 8-Story Timber | 8-Story Concrete | 12-Story Timber | 12-Story Concrete | 18-Story Timber | 18-Story Concrete |
|---|---|---|---|---|---|---|---|---|
| Exterior Walls | Aluminum stud | kg | - | 740 | - | 1562 | - | 2411 |
| | CLT | m$^3$ | 172 | - | 365 | - | 564 | - |
| | 3-5/8" Fiberglass mat | m$^2$ | - | 1638 | - | 3479 | - | 5387 |
| | 5/8" Gypsum board | m$^2$ | 1638 | 1638 | 3479 | 3479 | 5387 | 5387 |
| | 5" Mineral wool | m$^2$ | 1638 | - | 3479 | - | 5387 | - |
| | 1-1/2" Polystyrene | m$^2$ | - | 1638 | - | 3479 | - | 5387 |
| Floors | 3/8" Acoustic mat | m$^2$ | 8271 | - | 12,095 | - | 18,317 | - |
| | CLT | m$^3$ | 1444 | - | 2112 | - | 3199 | - |
| | Concrete | m$^3$ | 372 | 2293 | - | 2865 | 374 | 4666 |
| | Gypsum concrete | m$^3$ | 420 | - | 614 | 1537 | 931 | - |
| | 3/8" PE vapor barrier | m$^2$ | 1220 | 1214 | 1533 | - | 1227 | 1229 |
| | Rebar | kg | 5401 | 41,697 | - | 54,551 | 5,444 | 87,245 |
| Foundation | Concrete | m$^3$ | 458 | 674 | 1402 | 1874 | 602 | 818 |
| | Rebar | kg | 23,981 | 20,366 | 102,676 | 187,233 | 28,755 | 34,696 |
| Interior Walls | CLT | m$^3$ | 461 | - | 801 | - | - | - |
| | Concrete | m$^3$ | - | 516 | - | 961 | 1479 | 1419 |
| | 3-5/8" Fiberglass mat | m$^2$ | 3375 | 3511 | 5774 | 5934 | 8506 | 8830 |
| | 5/8" Gypsum board | m$^2$ | 29,566 | 24,041 | 84,107 | 40,707 | 148,456 | 60,959 |
| | 5-1/2" Mineral wool | m$^2$ | - | - | - | - | - | - |
| | Rebar | kg | - | 73,125 | - | 145,244 | - | 214,477 |
| | Steel stud | kg | 15,756 | 16,302 | 25,350 | 25,974 | 37,674 | 38,844 |
| | Exterior Brace Framing | kg | 10,534 | - | 16,272 | - | 24,369 | - |

**Table 3.** Bill of materials for 8-, 12-, and 18-story mass timber and concrete designs in the Northeast.

| Assembly | Material | Unit | 8-Story Timber | 8-Story Concrete | 12-Story Timber | 12-Story Concrete | 18-Story Timber | 18-Story Concrete |
|---|---|---|---|---|---|---|---|---|
| Columns and Beams | Concrete | m$^3$ | - | 93 | - | 385 | - | 586 |
| | Glulam | m$^3$ | 758 | - | 1207 | - | 1733 | - |
| | Rebar | kg | - | 59,392 | - | 139,608 | - | 285,609 |
| Exterior Walls | Aluminum stud | kg | - | 738 | - | 1550 | - | 2400 |
| | CLT | m$^3$ | 172 | - | 364 | - | 564 | - |
| | 3-5/8" Fiberglass mat | m$^2$ | - | 1638 | - | 3479 | - | 5387 |
| | 5/8" Gypsum board | m$^2$ | 1638 | 1638 | 3479 | 3479 | 5387 | 5387 |
| | 5" Mineral wool | m$^2$ | 1638 | - | 3479 | - | 5387 | |
| | 1-1/2" Polystyrene | m$^2$ | - | 1638 | - | 3479 | - | 5387 |
| Floors | 3/8" Acoustic mat | m$^2$ | 8229 | - | 12,053 | - | 18,277 | - |
| | CLT | m$^3$ | 1437 | - | 2105 | - | 3192 | - |
| | Concrete | m$^3$ | 372 | 2293 | - | 2865 | 370 | 6894 |
| | Gypsum concrete | m$^3$ | 418 | - | 612 | - | 928 | - |
| | 3/8" PE vapor barrier | m$^2$ | 1220 | 1214 | 1533 | 1537 | 1214 | 1229 |
| | Rebar | kg | 5401 | 41,697 | - | 54,551 | 5388 | 87,245 |
| Foundation | Concrete | m$^3$ | 367 | 674 | 1402 | 1874 | 601 | 818 |
| | Rebar | kg | 12,498 | 20,366 | 102,743 | 187,233 | 28,741 | 34,696 |
| Interior Walls | CLT | m$^3$ | 1152 | - | 2053 | - | 1835 | - |
| | Concrete | m$^3$ | - | 516 | - | 961 | 1474 | 1419 |
| | 3-5/8" Fiberglass mat | m$^2$ | 280 | 3511 | 500 | 5934 | 757 | 8830 |
| | 5/8" Gypsum board | m$^2$ | 13,724 | 24,041 | 74,381 | 40,707 | 175,872 | 60,959 |
| | 3" Mineral wool | m$^2$ | 3287 | - | 5617 | - | 8317 | - |
| | Rebar | kg | - | 73,125 | - | 145,206 | 202,066 | 214,477 |
| | Steel stud | kg | 8112 | 16,302 | 13,182 | 25,974 | 19,812 | 38,844 |
| | Exterior Brace Framing | kg | | | | | | |

**Table 4.** Bill of materials for 8-, 12-, and 18-story mass timber and concrete designs in the Southeast.

| Assembly | Material: Name | Unit | 8-Story | | 12-Story | | 18-Story | |
|---|---|---|---|---|---|---|---|---|
| | | | Timber | Concrete | Timber | Concrete | Timber | Concrete |
| Columns and Beams | Concrete | m³ | - | 93 | - | 385 | - | 586 |
| | Glulam | m³ | 752 | - | 1207 | - | 1733 | - |
| | Rebar | kg | - | 59,392 | - | 139,608 | - | 386,927 |
| Exterior Walls | Aluminum stud | kg | - | 730 | | 1534 | - | 2375 |
| | CLT | m³ | 172 | - | 364 | | 564 | |
| | 3-5/8" Fiberglass mat | m² | | 1638 | | 3479 | - | 5387 |
| | 5/8" Gypsum board | m² | 1638 | 1638 | 3479 | 3479 | 5387 | 5387 |
| | 5" Mineral wool | m² | 1638 | - | 3479 | - | 5387 | - |
| | 1-1/2" Polystyrene | m² | - | 1638 | - | 3,479 | - | 5387 |
| Floors | 3/8" Acoustic mat | m² | 8229 | - | 12,053 | | 18,277 | - |
| | CLT | m³ | 1437 | - | 2105 | - | 3192 | |
| | Concrete | m³ | 372 | 2293 | - | 2865 | 370 | 4666 |
| | Gypsum concrete | m³ | 418 | - | 612 | - | 928 | - |
| | 3/8" PE vapor barrier | m² | 1218 | 1214 | 1533 | 1537 | 18,277 | 1229 |
| | Rebar | kg | 5401 | 41,697 | - | 54,551 | 8364 | 87,245 |
| Foundation | Concrete | m³ | 367 | 674 | 1402 | 1874 | 601 | 818 |
| | Rebar | kg | 12,498 | 20,366 | 102,743 | 187,233 | 28,741 | 34,699 |
| Interior Walls | CLT | m³ | 1153 | - | 2053 | - | 1835 | - |
| | Concrete | m³ | - | 516 | - | 961 | 1474 | 1419 |
| | 3-5/8" Fiberglass mat | m² | 280 | 323 | 500 | 5934 | 757 | 8830 |
| | 5/8" Gypsum board | m² | 13,724 | 24,041 | 74,381 | 40,707 | 175,872 | 60,959 |
| | 5-1/2" Mineral wool | m² | 3287 | - | 5617 | - | 8317 | |
| | Rebar | kg | - | 73,125 | - | 145,206 | 202,066 | 214,477 |
| | Steel stud | kg | 8112 | 16,302 | 13,300 | 25,974 | 19,812 | 38,844 |
| | Exterior Brace Framing | kg | | | | | | |

*2.3. Whole-Building Life-Cycle Assessment*

The WBLCA follows ISO 21931 and EN 15978 standards [14,15], which define the building life-cycle stages from products stage (A), sse stage (B) to end-of-life stage (C) and beyond system stage (D), as depicted in Figure 3.

The framework of WBLCA is similar to the general LCA framework as defined in ISO 14040:2006 [17]. Figure 4 shows the WBLCA framework stages, and details of each stage for this study are described in the following sections.

2.3.1. Goal and Scope

The scope of this WBLCA is to conduct a cradle-to-gate analysis (from the acquisition of the raw material until building construction) and identify differences in the impact of building materials, transportation, and construction between mass timber and concrete designs. The goal of this study is to evaluate the environmental impacts of three mass timber buildings and compare them with those of functionally equivalent concrete buildings within each region, as described in Section 2.1.

2.3.2. Reference Unit

The reference unit of this WBLCA is defined as "providing 1 m² of living floor area in the building designed for the U.S. regions".

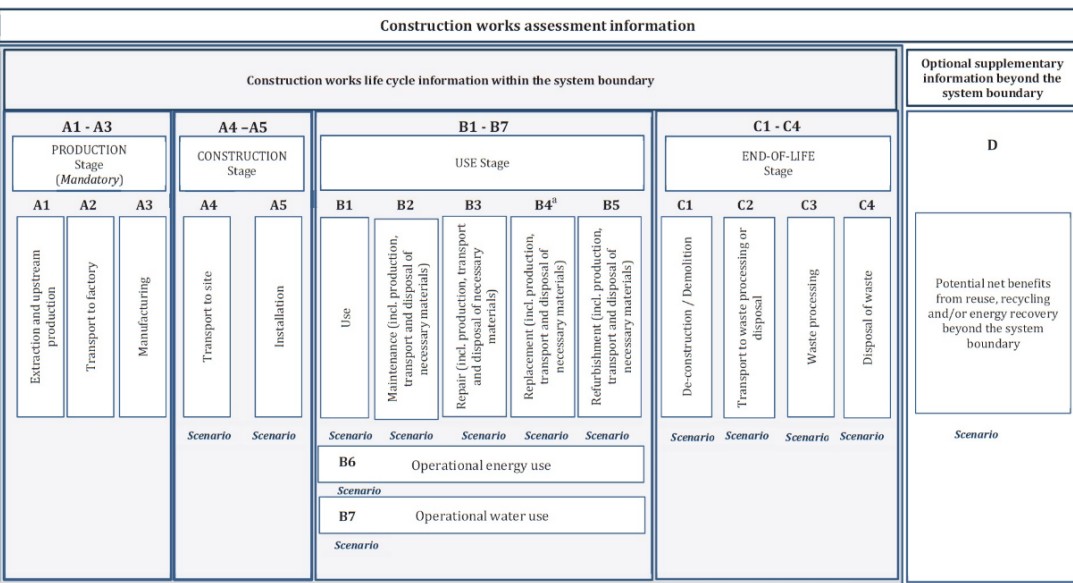

**Figure 3.** Building life-cycle stages and their information modules for construction products and construction works—from ISO 21931.

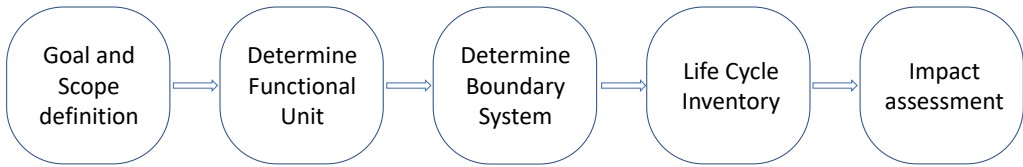

**Figure 4.** Whole Building LCA stages.

### 2.3.3. System Boundary

The system boundary defines which life-cycle phases are included in the analysis. The various processes that occur at each stage of a building life cycle are classified and grouped in "modules", labeled as A1–C4, as shown in Figure 3.

The system boundary for this WBLCA study is cradle to gate and includes modules A1—resource extraction, A2—transportation of materials to product manufacturing, A3—product manufacturing, A4—transportation of building products to construction site, and A5—building installation. Excluded from the study are life-cycle stages B, C, and D.

### 2.3.4. Life-Cycle Inventory

The life-cycle inventory of a product includes the collected primary data for each unit process of making the product, secondary data from established databases to quantify the raw material, and energy inputs and emission outputs of the manufactured product. The inventory analysis includes data collection, calculation, and allocation between co-products or product systems, and accounting for biogenic carbon uptake and emissions, carbonation, and more.

The life-cycle inventory of a building includes the collection of materials and quantity, selected LCI or EPDs for the building products, and estimated material transportation within the system boundary.

The life-cycle inventory (LCI) datasets for the collected building materials from all the building designs in this study are drawn from the DATASMART Package, which is composed of the US LCI database and US-Ecoinvent database. Cross-laminated timber and glulam are modeled based on manufacturing data in Chen et al. [18], Huang et al. [19], and Bowers et al. [20] for the three regions. The selected LCIs for each building material are listed in Section 3.1.2.

2.3.5. Life-Cycle Impact Assessment

The life-cycle impact assessment is a methodology that incorporates the inventory data of a product or system into the defined environmental impact categories and assesses the total environmental impacts associated with all stages of a product's life, from raw material extraction through material processing, manufacturing, use, repair and maintenance, and disposal or recycling. Conducting the LCA of a product or system is aimed to evaluate its overall impact and implement strategies to improve its environmental performance. LCA is a standardized methodology to evaluate various environmental impacts of products and services following the ISO standard. The LCA impact indicators for a building are defined by the ISO standard 21930 [21]. These impact indicators can be calculated through different methodologies developed around the world and approved by international research committees, thus incorporated in the LCA software for applications. For example, these methods include TRACI, CML, CED, and others that are embedded in the SimaPro software. TRACI is a commonly used LCA method in North America because it was developed for North American product impact assessment. A summary of the environmental indicators reported in this study are presented in the Table 5.

**Table 5.** Selected impact indicators reported.

| Reporting Category by ISO 21930:2017 | Indicator Name | Abbreviation | Units | Method |
|---|---|---|---|---|
| Core Mandatory Impact Indicators | Global warming potential, fossil | GWP | kg CO2e | TRACI |
| Core Mandatory Impact Indicators | Depletion potential of the stratospheric ozone layer | ODP | kg CFC11e | TRACI |
| Core Mandatory Impact Indicators | Acidification potential of soil and water sources | AP | kg SO2e | TRACI |
| Core Mandatory Impact Indicators | Eutrophication potential | EP | kg PO4e | TRACI |
| Core Mandatory Impact Indicators | Formation potential of tropospheric ozone | SFP | kg O3e | TRACI |
| Core Mandatory Impact Indicators | Abiotic depletion potential (ADP fossil) for fossil resources; | ADPf | MJ, NCV | CML |
| Core Mandatory Impact Indicators | Abiotic depletion potential (ADP element) for fossil resources; | ADPe | kg, Sbe | CML |
| Core Mandatory Impact Indicators | Fossil fuel depletion | FFD | MJ Suplus | TRACI |
| Use of Primary Resources | Renewable primary energy carrier used as energy | RPRE | MJ, NCV | CED |
| Use of Primary Resources | Nonrenewable primary energy carrier used as energy | NRPRE | MJ, NCV | CED |
| Mandatory Inventory Parameters | Consumption of freshwater resources | FW | m3 | Manual from LCI |
| Indicators Describing Waste | Hazardous waste disposed | HWD | kg | Manual |
| Indicators Describing Waste | Nonhazardous waste disposed | NHWD | kg | Manual |

**Table 5.** *Cont.*

| Reporting Category by ISO 21930:2017 | Indicator Name | Abbreviation | Units | Method |
|---|---|---|---|---|
| Additional Inventory Parameters for Transparency | Biogenic Carbon Removal from Product | BCRP | kg CO2e | Manual |
| Additional Inventory Parameters for Transparency | Biogenic Carbon Emission from Product | BCEP | kg CO2e | Manual |
| Additional Inventory Parameters for Transparency | Biogenic Carbon Emission from Combustion of Waste from Renewable Sources Used in Production Processes | BCEW | kg CO2e | Manual from LCI |
| Additional Inventory Parameters for Transparency | Calcination Carbon Emissions | CCE | kg CO2e | Manual |

Results of LCA impacts from life-cycle stages A1–A5 in these indicators for all 18 buildings are presented in the next paper of this series [16], which highlights the differences between mass timber and concrete buildings and regional differences. This paper focuses only on modeling WBLCAs developed using the LCA software SimaPro 9.1 and various databases included.

**3. Discussion**

*3.1. Life-Cycle Inventory for Building Material, Energy, and Fuel Used in the Building Structures*

3.1.1. Data Quality for the Prospective LCI/LCA

- Precision: All LCI/LCA data sources used were compiled in accordance with ISO 14040/14044 procedures and requirements [17,22];
- Consistency: The assessment draws from several databases (Table 3) that are consistent with the system boundary and scope;
- Reproducibility: Most LCI data used is publicly available or referenced to particular data sets such that reproducibility is possible.

Representativeness: Recent and regional LCI data sets (especially for CLT and glulam) were used and fall within the EN 15978 limit of ten years. Geographical coverage was for North America and is representative of the region (PNW, NE, and SE) where the buildings are located. Technological coverage reflects the physical reality of the products found in the building.

3.1.2. The Prospective LCIs for Each Material Designed in the Mass Timber and Concrete Buildings

The prospective LCIs for each material designed in the mass timber and concrete buildings are presented in Table 6.

**Table 6.** Summary of the material, energy, and fuel used with their associated LCI data sources used in both mass timber and concrete buildings designed for the comparative study.

| Material/Energy/Fuel | LCI Process | Database/Data Source |
|---|---|---|
| Acoustic panel | Gypsum plaster board, at plant/US US-EI U | US-EI 2.2 |
| Aluminum stud | Galvanized steel sheet, at plant NREL/RNA U | US-EI 2.2 |
| CLT | CLT | [18,19] |
| Concrete | Concrete, 3000 psi | Athena impact estimator |

**Table 6.** *Cont.*

| Material/Energy/Fuel | LCI Process | Database/Data Source |
|---|---|---|
| Concrete | Concrete, 4000 psi | Athena impact estimator |
| Concrete | Concrete, 5000 psi | Athena impact estimator |
| Construction energy | Diesel, combusted in industrial equipment NREL/US U | US-EI 2.2 |
| Electricity | Regional Grids WECC, NPCC, WECC | Ecoinvent |
| Exterior Brace Framing | Hot rolled sheet, steel, at plant NREL/RNA U | US-EI 2.2 |
| Glulam | Glulam | CORRIM and [19] |
| Gypsum concrete | Proxy process | US-EI 2.2 |
| Gypsum wallboard | Gypsum fibre board, at plant/US US-EI U | US-EI 2.2 |
| Insulation | Glass wool mat, at plant/US US-EI U | US-EI 2.2 |
| Insulation | Rock wool, packed, at plant/US US-EI U | US-EI 2.2 |
| Insulation | Polystyrene, extruded (XPS), at plant/US- US-EI U | US-EI 2.2 |
| Polystyrene Insulated Sheathing | Polystyrene, extruded (XPS), at plant/US- US-EI U | US-EI 2.2 |
| Rail Transport | Transport, train, diesel powered NREL/US U | US-EI 2.2 |
| Rebar | Reinforcing steel, at plant/US- US-EI U | US-EI 2.2 |
| Rectangular Mullion: 3–5/8″ C Stud | Galvanized steel sheet, at plant NREL/RNA U | US-EI 2.2 |
| Road Transport | Transport, combination truck, diesel powered NREL/US U | US-EI 2.2 |

### 3.2. Building LCA Assumptions

#### 3.2.1. Building Site Assumptions

Because this study was designed to evaluate the broad geographic regional impacts with mass timber products penetrating the building sector, the buildings were not designed with any particular site in mind. However, for some structural and LCA analyses, certain site assumptions had to be made given the need for appropriate soil pressure, seismic condition, transportation of products, and material sourcing for the mass timber products. Therefore, the exact locations for the three regional buildings were chosen, as shown in Figure 2.

Although the sites from different weather zones presented potential humidity and temperature effects on the mass timber's service life, durability, and safety in construction [23,24] that may affect the life-cycle assessment results, it was not examined in this study. It would be important for future studies to look into such moisture effects on mass timber buildings' overall performance and WBLCA results.

#### 3.2.2. Material Assumptions

Because the focus of this study was to evaluate the impact of regional mass timber use, the following (Table 7) manufacturing facility and wood species were chosen to represent locally supplied materials and manufacturing.

**Table 7.** Assumptions used for mass timber production sites and species.

| Geographic Regions | Mass Timber Production | Species |
|---|---|---|
| Northeast (NE) | Lincoln, Maine | Eastern Spruce and White Pine |
| Pacific Northwest (PNW) | Seattle, Washington | Douglas-fir, Western Hemlock |
| Southeast (SE) | Dothan, Alabama | Southern Pine |

Because building designs in this study are preliminary designs without checking details for wind and lateral drift considerations, they are only to meet assumptions for gravity loading. Thus, all superstructure concrete would have a minimum specified compressive strength of f'c1 = 5000 psi (Table 8); particularly, post-tensioning would likely be used in the concrete building floors. All other specified concrete compressive strengths for different building components are also shown in Table 8.

**Table 8.** Concrete compressive strength assumptions for all the building components in mass timber and concrete designs.

| Category | Concrete Compressive Strength (psi) |
|---|---|
| Below Grade Foundation Walls | 4000 |
| Grade Beams | 4000 |
| Slabs-on-grade | 4000 |
| Pile caps | 4000 |
| Spread footings | 3000 |
| Superstructure | |
| Floors | 5000 |
| Columns | 5000 |
| Shaft walls | 5000 |

### 3.2.3. Structural Assumptions

Each foundation system was designed for the conceptual design level of building loads for each building material, mass timber or concrete, as well as for conceptual level seismic/wind factors for each location. Table 9 shows that a spread footing foundation is used for the 8-story buildings and that a mat footing foundation option is used for the 12-story buildings. The 18-story superstructures must be supported on piles in addition to a spread footing foundation (Table 9). The heavier concrete superstructures required larger volumes of concrete and reinforcing steel in the foundations than the lighter timber superstructures in every pairing.

**Table 9.** Foundation type for each building design for all regions.

| Stories | Building Design | |
|---|---|---|
| | Mass Timber | Concrete |
| 8 | Spread Footing | Spread Footing |
| 12 | Mat Footing | Mat Footing |
| 18 | Piles and Spread Footing | Piles and Spread Footing |

### 3.2.4. Transportation Assumptions

The transportation of materials from manufacturers or distribution locations to the building site occurs in the A4 phase. Table 10 shows the distance of each material to the building sites for each location. All materials were assumed to be produced and sold domestically, and therefore only road and rail transportation modes were considered. For distances shorter than 500 miles (805 km), the materials were assumed to be transported by truck; for distances longer than 500 miles, they were assumed to use a combination of truck and rail transport.

### 3.2.5. Construction and Installation Assumptions

The construction and installation phase, referred to as the A5 module in the life-cycle assessment (EN 15978/ISO 21931), considers all ground and onsite works for building erection, heating/cooling and ventilation provided, and onsite water and waste management, among others. However, we calculated only diesel fuel consumption for lifting the building materials by crane as one estimate of energy use for this A5 stage for LCA impacts. The fuel use in liters (L) for construction was calculated based on the amount of material used in each building design, and by assuming that the materials were lifted by crane to half the height of the building using the following equation [25]:

$$Fuel(L) = 0.000037Mh + \frac{M}{500} + 0.83 \tag{1}$$

where:

*M = mass of the material being lifted in kg*
*h = height at which the material is being lifted. Half of the building height was assumed.*

The total diesel fuel used for each building construction is summarized in Table 11 for all three regions.

**Table 10.** Transportation Assumptions (km) of Building Materials to Building Site in the three regions.

| Material Name | PNW | | NE | | SE | |
|---|---|---|---|---|---|---|
| | Truck | Rail | Truck | Rail | Truck | Rail |
| | **Kilometer (Km)** | | | | | |
| 25 ga steel, Galvanized steel sheet | 30 | 1321 | 16 | 918 | 16 | 1151 |
| Acoustic Mat, Gypsum plaster board | 219 | - | 391 | | 41 | |
| Aluminum,3-5/8″ Metal Stud | 30 | 1321 | 292 | | 16 | 1371 |
| CLT | 473 | - | 460 | | 354 | |
| Concrete | 52 | - | 17 | | 9 | |
| Exterior Brace Framing | 87 | 1321 | - | - | - | |
| Extruded Polystyrene Insulated Sheathing | 285 | - | 98 | | 42 | |
| Fiberglass Batt, Glass wool mat | 16 | 1321 | 98 | | 42 | |
| Glulam | 490 | - | 460 | | 332 | |
| GYP, Gypsum wallboard | 219 | - | 391 | | 27 | |
| Gypsum concrete | 219 | - | 391 | | 41 | |
| Mineral Wool, Rock wool | 285 | - | 98 | | 42 | |
| Polyethylene film membrane | 16 | 2253 | 637 | | 82 | |
| Rebar | 32 | - | 53 | | 31 | |

**Table 11.** Summarized fuel use for each building construction during the A5 stage.

| Region | Diesel Use (L) | | | | | |
|---|---|---|---|---|---|---|
| | Timber | Concrete | Timber | Concrete | Timber | Concrete |
| | **8** | | **12** | | **18** | |
| PNW | 11,856 | 22,481 | 23,519 | 44,990 | 42,835 | 64,171 |
| NE | 10,943 | 21,677 | 23,585 | 43,517 | 43,726 | 61,842 |
| SE | 11,997 | 21,639 | 25,574 | 43,424 | 46,819 | 62,270 |

*3.3. Comparative Building LCA*

The purpose of conducting WBLCAs is to make appropriate comparisons for effective material selections and efficient building designs to lower the whole-building environmental impacts. For comparative building LCA, "apples to apples" comparison is very important. Buildings include a multitude of mass timber, steel, concrete, hybrid structures, and various design options in accordance with the regional building codes. To perform a fair comparison, the team designed every pair of buildings (mass timber and concrete) according to the standards and building codes.

3.3.1. Functional Equivalents

This study closely follows EN 15978, which requires the identification of a functional equivalent for the building to enable a valid basis for future comparisons to other buildings. According to EN 15978, a functional equivalent is "the quantified functional requirements and/or technical requirements for a building or an assembled system (part of works) for use as a basis for comparison." In other words, the functional equivalent is a set of design criteria that both buildings must have in common to ensure an apples-to-apples comparison.

### 3.3.2. Comparable Building Designs

Three pairs of model buildings were designed for three U.S. regions to conform to three tall mass timber building types established in the 2021 International Building Code (ICC) revisions. The results allow for comparisons among regions, each of which has different energy mixes and timber species, and, in the case of the PNW, additional seismic considerations that drive differences in the building design. In total, 18 different modeling conditions were assessed (3 heights × 3 regions × 2 building materials). Conventional buildings were designed for functionally equivalent comparisons with each mass timber buildings (e.g., equivalent height, interior floor space, interior divisions, and intended uses).

### 3.4. Contribution Analysis from WBLCA

Examining the contributions from building materials and building assemblies can help to identify the hotspots of global warming potential (GWP) contributions and other environmental impacts. Such contribution analysis in the WBLCA can help researchers and policymakers focus on the identified causal factors required to reduce the impacts. This project analyzed the GWP contributions from all the building materials and from all the building assemblies for each of the building designs. As expected, concrete and rebar are the dominant contributors to GWP in all concrete buildings and are significant contributors in the mass timber buildings because of their inherent intensive carbon footprints from the extracting and manufacturing processes. Therefore, reducing the use of these products or substituting these products with lower carbon footprint products such as mass timber is of interest for policymakers and architecture designers. More gypsum board is applied to the mass timber buildings as height increases from 8 to 12 to 18 stories because of the building code requirement. The ICC code specifies the use of gypsum wall board (GWB) as the requisite fire protection for noncombustible material such as wood for the wall assemblies. Codes require two to three layers of GWB over a percentage of exposed wood surfaces in the 12- and 18-story mass timber buildings. This raises the need for further investigation into the fire protection requirements for mid- to high-rise mass timber buildings and the equivalent comparable concrete designs. More detailed discussion and results from this research are presented in the next paper of the series [16].

### 3.5. Carbon Storage in Mass Timber Buildings

Wood products, such as mass timber, store a significant amount of carbon during their lifespan. Carbon storage in the building is calculated based on the quantity and species of the wood material used in the building. Storage time depends on the product's lifetime and its end-of-life fate. The longer the product is used, the longer the carbon stays in its storage form [26] until it is landfilled or burned. Such delayed emissions are especially important for mass timber products such as CLT and glulam beams because the functional life of such products can be much longer than other traditional wood products. In all the mass timber buildings designed in this study, carbon storage is greater than the carbon released (including both fossil and biogenic carbon) from the product's manufacturing process. Details of carbon storage in the design of mass timber buildings can be found in [27]. The temporal carbon storage benefits were calculated in this study using the cumulative radiative forcing (RF) integrated from the beginning of the time horizon to the last year of storage. The net global warming potential was calculated by subtracting the carbon storage benefit (expressed in terms of $CO_2$ eq/$m^2$) from the building's total GWP estimated from the WBLCA analysis for life-cycle stages A1–A5. From this temporal analysis, for all the mass timber buildings in all regions, the net GWP at year 80 (average lifespan of the building) was net negative, meaning that the 80-year carbon storage benefit more than offset the GWP from fossil greenhouse emissions, and the mass timber building is a carbon sink. The shorter the lifespan of the building, the lower the carbon sink. Another paper in this series [28] shows details of net carbon storage benefits for each of these mass timber buildings for a 100-year time horizon.

## 4. Conclusions and Recommendation

Comparative WBLCA is an objective tool for selecting building materials and designing buildings efficiently to lower total environmental impacts. Comparative LCAs are based on individual LCAs completed on functional equivalent products or buildings. The methodology used in this series of WBLCAs follows the ISO 21931 and EN 53978 standards. Eighteen WBLCAs for eighteen designed buildings were modeled, selecting LCIs for building materials from the US LCI database, Athena database, and US-Ecoinvent database. WBLCAs in this study were conducted in SimaPro LCA software, rather than using a commercial WBLCA tool. This made the LCA analysis more robust, transparent, and flexible in applying updated LCI datasets and specific material densities. Series papers [16,28], including this one, contain the designs, process, and results for the three selected U.S. regions: Seattle, Washington (PNW), Boston, Massachusetts (NE), and Atlanta, Georgia (SE). Building designs and assumptions were developed by atelierjones llc with contributions from the USDA Forest Products Laboratory and WoodWorks. The building LCAs were completed by teams at the University of Washington (PNW), USDA Forest Products Laboratory (NE), and the Consortium for Research on Renewable Industrial Materials (CORRIM) (SE). Together, the team developed a series of detailed, comparative life-cycle assessments of pairs of buildings using both mass timber and conventional materials. Results highlight the importance of building design, supply chain and manufacturing emissions, carbon storage, and the climate impact of each building type and region.

The scientific rigor in conducting WBLCAs sets the accuracy of the results. Selecting the product and material LCI from various databases is critical to the LCA results when no standardized building and material database is available for use. Transparency of these WBLCAs is even more important to the community for comparison and adoption. The transparent and detailed description of this series of WBLCAs is intended to provide an example and set the accountability of the conclusive arguments we derived from this study. Opportunities for improving the use of wood as a building material are endless, including improving choices of materials, building designs, innovative products, and building codes, including the use of mass timber for high-rise buildings that can store more carbon and displace fossil-intensive alternatives. A plethora of data exists on the favorable environmental performance of wood as a building material and energy source, and many opportunities appear for research to improve on current practices. Extending the service life of building materials with reuse and recyclability will further help to minimize the environmental impacts [29]. Future studies on the reuse of mass timber products will be a focus in the building sector.

**Author Contributions:** Conceptualization, H.G., S.L., M.P., F.P., I.G., M.W., C.C., S.J. and I.M.; methodology, H.G., S.L., M.P., F.P., I.G., C.C., S.J. and I.M.; validation, H.G., S.L., M.P., F.P., C.C., S.J. and I.M.; formal analysis, M.P., F.P., C.C., S.L. and H.G.; investigation, M.P., F.P., C.C., S.L. and H.G.; data curation, M.P., F.P., C.C., I.M. and S.L.; writing—original draft preparation, H.G.; writing—review and editing, S.L., M.P., F.P., C.C. and R.P.; supervision, H.G., M.P., I.G. and R.P.; project administration, M.W. and R.P.; funding acquisition, R.P. All authors have read and agreed to the published version of the manuscript.

**Funding:** This research was founded by The Climate and Land Use Alliance, the Doris Duke Charitable Foundation (DDCF), and the USDA Forest Service, Forest Products Laboratory, Forest Products Marketing Unit, grant number [17-CA-11111169-031].

**Institutional Review Board Statement:** Not applicable.

**Informed Consent Statement:** Not applicable.

**Data Availability Statement:** Not applicable.

**Acknowledgments:** This article makes up part of a larger 5-phased project which was initiated by The Nature Conservancy (nature.org) through generous support from The Climate and Land Use Alliance, and the Doris Duke Charitable Foundation (DDCF). The work upon which this project is based was also funded in whole or in part through a cooperative agreement with the USDA Forest

Service, Forest Products Laboratory, Forest Products Marketing Unit (17-CA-11111169-031) *. The Nature Conservancy initiated this project to further our collective understanding of the potential benefits and risks of increasing demand for forest products and ensuring that any increases are sustainable. The Conservancy's objectives are focused on delivering critical safeguard frameworks to mitigate any potential risks on forest ecosystems as mass timber demand increases. *In accordance with Federal Law and U.S. Department of Agriculture policy, this institution is prohibited from discriminating on the basis of race, color, national origin, sex, age, or disability. (Not all prohibited bases apply to all programs.) To file a complaint of discrimination, write USDA, Director, Office of Civil Rights, Room 326-W, Whitten Building, 1400 Independence Avenue, SW, Washington, DC 20250-9410 or call (202) 720-5964 (voice and TDD). USDA is an equal opportunity provider, employer, and lender.

**Conflicts of Interest:** The authors declare no conflict of interest. The funders had no role in the design of the study; in the collection, analyses, or interpretation of data; in the writing of the manuscript, or in the decision to publish the results.

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
