# Peer review of "Mass Timber Building Life Cycle Assessment Methodology for the U.S. Regional Case Studies"

_sustainability, doi:10.3390/su132414034_

Round 1

Reviewer 1 Report

The article is correctly conceived and even indicates the current use of wood as a construction material. Although it does not bring any news from the aspect of science, I recommend that it be published as an economic, environmental contribution to the understanding of the use of wood. In any case, a useful warning for the sustainable development of construction and the future of purposeful use of wood.

Author Response

This paper is part of the series paper on the large project organized by TNC with multiple institutions. Although this paper does not present or discuss any results from the comparative Whole Building LCAs, it lay out the method used and assumptions made for developing the LCA models. This paper also provided some guidelines for conducting whole building LCA comparative analysis based on the ISO and EN standards. It needs to be published together with other papers in the same series and submitted to this special Sustainability Issue. 

Reviewer 2 Report

Reviewer’s comments and suggestions

The manuscript presents a methodology for the  comparative life-cycle assessments of pairs of buildings using both mass timber and conventional materials in three different U.S. regional areas. In particular, the Whole-Building LCA methodology is adopted and a large amount of data is collected on the favorable environmental performance of wood as a building material and energy source. Natural climate solutions based on wood can indeed increase carbon storages or avoid greenhouse gas emissions. WBLCAs for 18 designed buildings were modeled selecting Life Cycle Inventories for 367 building materials from several databases. This allowed to make the LCA analysis more robust, transparent, and flexible. The results pointed out the importance of building design, manufacturing emissions, carbon storage, and the climate impact of each building type and region.

Whereas the large amount of information provided in the manuscript is very useful for designers and researchers, my main concern is that the climate impact is handled with a general approach, considering for example the different wood species in the studied regional areas, and LCI available data. On the contrary, it seems that the effects on the moisture on wood products is not considered. In my comments in the reviewed pdf file, I have given several suggestions to include some considerations on the importance of the moisture variations in the hygroscopic wood that can affect the safety, serviceability and durability of timber components and buildings. The Authors could think to integrate more information on the impact of moisture in wood in their methodology in future studies. Furthermore, possible activities of reuse of wood components could be also highlighted and considered for future studies within the WBLCA methodology. Some additional references are suggested below.

In addition, some suggestions to improve the presentation of figure and tables are given in the reviewed pdf file.

Yours sincerely,

The Reviewer

Additional References

  • Frühwald E, Serrano E, Toratti T, Emilsson A, Thelandersson S. Design of safe timber structures – how can we learn from structural failures in concrete, steel and timber? Research report TVBK-3053. Lund University; 2008. ISSN: 0349- 496.
  • Brischke, C.; Alfredsen, G.; Humar, M.; Conti, E.; Cookson, L.; Emmerich, L.; Flæte, P.O.; Fortino, S.; Francis, L.; Hundhausen, U.; et al. Modelling the Material Resistance of Wood—Part 3: Relative Resistance in above and in Ground Situations— Results of a Global Survey. Forests 12, 590 (2021). https://doi.org/ 10.3390/f12050590
  • Hradil, P., Talja, A., Wahlström, M., Huuhka, S., Lahdensivu, J., & Pikkuvirta, J. (2014). Re-use of structural elements: Environmentally efficient recovery of building components.

Author Response

Thank you for this insightful review. The reviewer has very good knowledge in the Mass timber buildings and LCA works. Appreciate the comments and constructive reviews. We have taken all the considerations and updates in the paper according to this review. See the individual response to each comment in the document (attached). 

thanks for providing more references. There are added in the paper.
